# DEITalk: Speech-Driven 3D Facial Animation with Dynamic Emotional Intensity Modeling

## ABSTRACT

Speech-driven 3D facial animation aims to synthesize 3D talking head animations with precise lip movements and rich stylistic expressions. However, existing methods exhibit two limitations: 1) they mostly focused on emotionless facial animation modeling, neglecting the importance of emotional expression, due to the lack of high-quality 3D emotional talking head datasets, and 2) several latest works treated emotional intensity as a global controllable parameter, akin to emotional or speaker style, leading to over-smoothed emotional expressions in their outcomes. To address these challenges, we first collect a 3D talking head dataset comprising five emotional styles with a set of coefficients based on the MetaHuman character model and then propose an end-to-end deep neural network, **DEITalk**, which conditions on speech and emotional style labels to generate realistic facial animation with dynamic expressions. To model emotional saliency variations in long-term audio contexts, we design a dynamic emotional intensity (DEI) modeling module and a dynamic positional encoding (DPE) strategy. The former extracts implicit representations of emotional intensity from speech features and utilizes them as local (high temporal frequency) emotional supervision, whereas the latter offers abilities to generalize to longer speech sequences. Moreover, we introduce an emotion-guided feature fusion decoder and a four-way loss function to generate emotion-enhanced 3D facial animation with controllable emotional styles. Extensive experimental results demonstrate that our method outperforms existing state-of-the-art methods. We recommend watching the video demo provided in our supplementary material for detailed results.

## CCS CONCEPTS

• **Computing methodologies** → **Computer vision**.

## KEYWORDS

facial animation, emotion, audio-to-face, deep learning

## 1 INTRODUCTION

Speech-driven 3D facial animation aims to synthesize realistic facial movements of 3D characters based on arbitrary speech input. This capability has significant applications in various industries such as film production, video games, and virtual reality [9, 32, 38, 46]. Thus, it has garnered significant attention from both academia and

*ACM MM, 2024, Melbourne, Australia*
© 2024 Copyright held by the owner/author(s). Publication rights licensed to ACM.
ACM ISBN 978-x-xxxx-xxxx-x/YY/MM
https://doi.org/10.1145/nnnnnnn.nnnnnnn

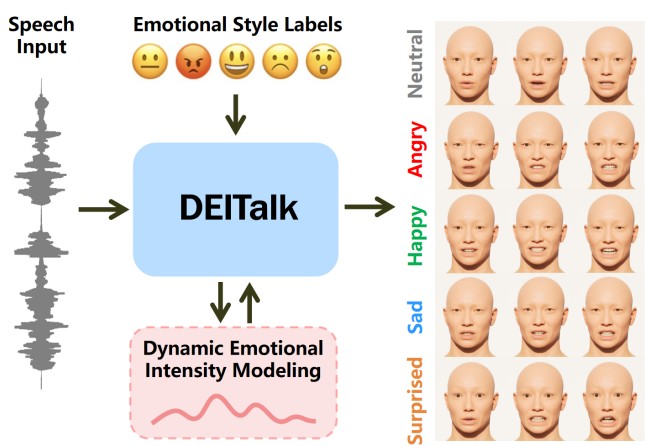

**Figure 1: Rendered results of our DEITalk. Given a speech sequence and an emotional style label as input, our DEITalk outputs realistic 3D facial animation sequences with corresponding emotional expressions, where the intensity of the expressions varies over time.**

industry recently. Learning-based methods [5, 11, 15] for speech-driven 3D facial animation have become a focal point of current research. Compared to traditional methods that require professional animators to manually create animations based on a set of rules, these data-driven approaches can greatly reduce annotation and time costs while achieving realistic 3D character facial animations.

However, recent methods still suffer from two major limitations. First, due to the absence of suitable datasets, many speech-driven 3D efforts [6, 10, 31, 39] focused mainly on synthesizing highly synchronized lip movements, as well as reconstructing natural upper facial movements. Moreover, they neglect the significance of emotions in facial animation expression, leading to the results falling short of realism. Secondly, some works view emotional information as a style embedding and represent them in the form of one-hot vectors [16, 20, 37]. Although such a strategy provides an effective means of controlling emotional expression in animation, it tends to steer the learned results towards the mean of different emotional styles, i.e., losing personalized expression of emotions.

We argue that facial animation consists primarily of two distinct components: speech and emotional expression. Lip movements, driven by speech, represent a high-frequency phenomenon that requires precise synchronization with speech. Conversely, emotions can be characterized by two sub-components: emotional style, a consistent, low-frequency pattern, and emotional intensity, a relatively higher-frequency descriptor influenced by the level of emotional significance at various moments. However, estimating dynamic emotional intensity from audio or video using data-driven methods remains a challenging field. This challenge mainly arises

from the difficulty of constructing large-scale datasets containing emotional intensity annotations for audio or video content, which requires expert evaluation of data across temporal dimensions and incurs significant labor and time costs. Furthermore, emotional intensity perception is inherently subjective and complex, influenced by individual expression variances and cultural disparities, thus making it difficult to establish a unified standard for evaluation. Certain approaches treat emotional intensity as a global control condition, enabling emotional intensity editing [7, 28], while they may restrict the expressive range of the model for emotions.

To synthesize realistic emotional 3D talking head animation, we propose a novel speech-driven 3D facial animation method (shown in Figure 1) called DEITalk, where a dynamic emotional intensity modeling module is proposed as our key contribution (illustrated in Figure 2). This module learns a shared latent space for audio and visual modalities via contrastive learning, which can be used to model the saliency of emotions temporally, i.e., dynamic emotional intensity. In addition, we design an emotion-guided feature fusion decoder, in which multiple different types of features are decoded by a Temporal Convolutional Network (TCN) [23] and output 185 emotion-enhanced control rig coefficients for facial expression rendering. Furthermore, a dynamic positional encoding strategy and a diversified loss function are further proposed to better enhance the model's capability for emotional expression.

To address the lack of a publicly available 3D emotional talking head dataset, we adopt a combination to capture 3D facial movements: the MetaHuman character model [13], developed by Epic, and the Live Link Face application, available in the iOS App Store. Subsequently, 185 control rig coefficients are acquired through associated MetaHuman plugins, where each coefficient corresponds to a group of facial muscles. Using the aforementioned methods, we establish a novel 3D emotional talking head dataset, termed ETH-3D. In contrast to existing 3D scan datasets such as VOCASET [6] and BIWI [12], ETH-3D employs low-dimensional data (i.e., control rig) to depict rich and intricate facial motions. This data serves as a universal facial motion representation that can be generalized to any predefined MetaHuman character.

In summary, the primary contributions of our work include:

- First, we propose a novel approach, DEITalk, for speech-driven 3D facial animation, which enables the synthesis of realistic expressions with a specified emotional style while adapting to the dynamic emotional intensity in the speech.
- Second, we introduce a dynamic emotional intensity modeling module that employs a contrastive learning strategy to learn the correlation between speech and facial movements in emotional saliency. The module generates joint embeddings which are then transformed into dynamic emotional intensity features. These features guide the model in synthesizing convincing facial expression changes.
- Third, we capture a 3D emotional talking head dataset, ETH-3D, comprising five distinct emotional styles. The dataset includes audio-visual pairs that demonstrate reasonable dynamic variations in emotional intensity.
- Finally, we are among the pioneering developers of generation models based on the MetaHuman control rig. In comparison to mesh-based models, our approach not only

synthesizes more detailed and natural facial animations but is also better suited for integration into industrial pipelines.

## 2 RELATED WORK

### 2.1 Speech-Driven 3D Facial Animation

With the rapid development and widespread adoption of 3D games and virtual reality technologies, traditional speech-driven or image-driven 2D facial animation methods [4, 14, 17, 18, 34, 43, 44] cannot be directly applied to 3D character models. Consequently, speech-driven 3D facial animation has recently garnered significant attention [3, 5, 6, 10, 11, 15, 16, 19, 20, 27, 28, 31, 33, 35, 39, 45].

Most previous methods have significant limitations due to the lack of high-quality publicly available 3D datasets for various emotions. For instance, both VOCA [6] and Faceformer [10] are only capable of achieving decent lip movements but exhibit limited upper facial movements because they are trained on the VOCASET dataset [6], which mainly focuses on descriptions of lip movements. Although Meshtalk [31] enhances diversity in upper facial movements by learning a categorical latent space, it fails to convey different emotions. Substantially, the above methods focus solely on achieving different speaking styles among individuals for animation, controlled by inserting style vectors of training individuals, while overlooking an important factor that affects the perception of animation: emotion.

Recently, Emotalk [28] introduced an emotion disentanglement mechanism to extract emotional information from speech input. However, it relies on curated training data, limiting its effectiveness in extracting emotions in zero-shot settings. Furthermore, since it only offers control over the intensity of emotional expression rather than the emotional type, users face challenges in obtaining animations representing desired emotional categories. 3D-TalkEmo [37] utilizes emotional style labels to adjust facial animation's emotional expression. However, due to the lack of description regarding emotional expression intensity and treating emotional style labels as global control information, the resulting animations for the same emotional style are highly similar, lacking high-frequency details related to speech input.

In contrast to existing works, this paper conceptualizes the intensity of emotion as a high-frequency implicit learnable attribute that exhibits correlations with both speech and facial movements. Through the integration of this implicit attribute with explicit emotional styles controlled by users, the model is guided to generate more nuanced and intricate facial expressions.

### 2.2 Audiovisual Emotional Intensity Estimation

Unlike the significant advancements witnessed in recent years in deep learning models [26, 40] for speech emotion recognition (SER) tasks, estimating the intensity of emotional reactions in audio or video using data-driven approaches remains a highly challenging endeavor. This is primarily attributed to the inherently subjective nature of human perception regarding the intensity of emotional expression and the difficulty in establishing universally applicable quantitative assessment metrics for emotional intensity.

A task of emotion reaction intensity (ERI) estimation was proposed by the fifth Affective Behavior Analysis in the wild (ABAW) competition [21]. ABAW released an audio-visual dataset called

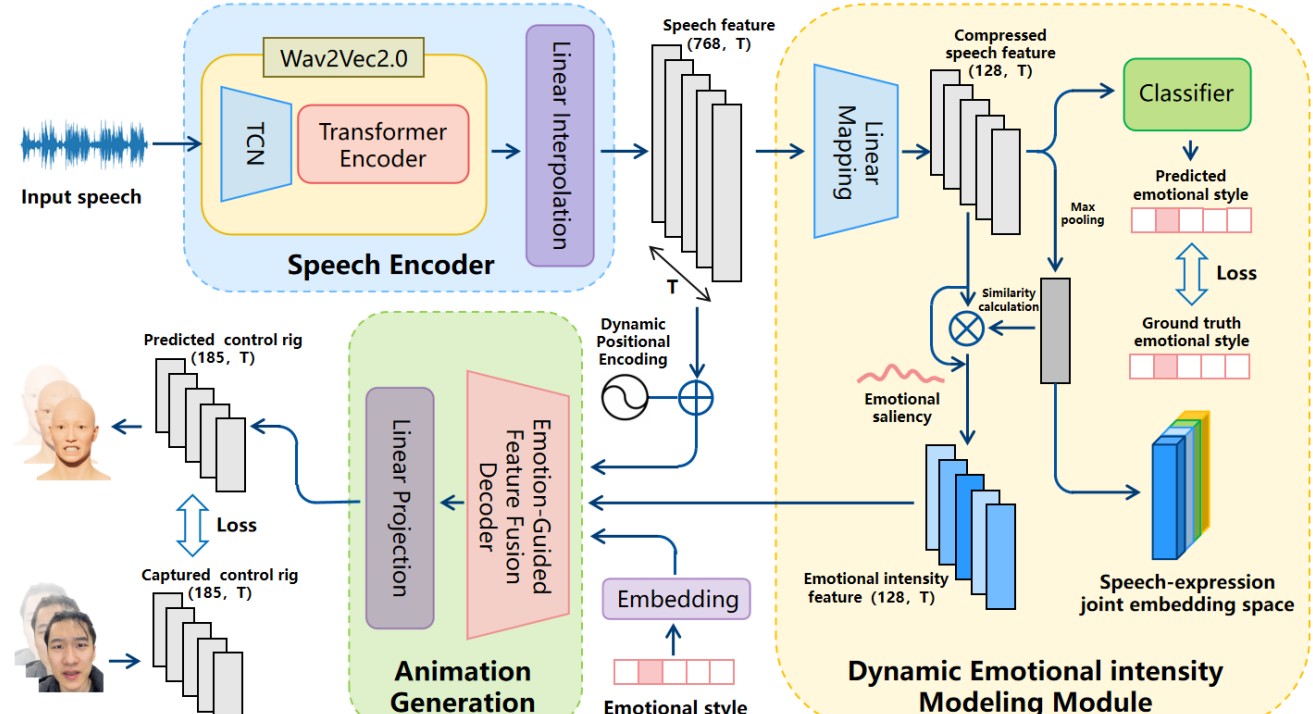

**Figure 2: Overview of our DEITalk, which takes a speech sequence $A_{1:T}$ and emotional style $e^{sty}$ as input. The speech sequence undergoes initial encoding using Wav2Vec 2.0, followed by a linear interpolation layer for speech feature resampling. Subsequently, the resampled speech feature is passed through the dynamic emotional intensity modeling module to extract emotional intensity features. Next, a dynamic positional encoding strategy is applied to the speech feature. Finally, the speech feature, along with the emotional intensity feature and the emotional style embedding, is decoded by an emotion-guided feature fusion decoder. This decoder outputs control rig coefficients, which can then be applied to manipulate MetaHuman characters, resulting in emotive facial animations.**

Hume-Reaction, where individuals' reactions to emotional stimuli are annotated across seven emotional expression intensities. In response, numerous deep learning models [22, 24, 29, 42] have been proposed for this ERI task. However, the emotional intensity annotations in the Hume-Reaction dataset are globally unique for each audiovisual sample. Such annotations may prove inadequate for analyzing long-context audiovisual data, thereby limiting the applicability of these methods within our particular domain.

In this paper, we regard emotional intensity as a continuous attribute evolving over time, which can be implicitly modeled. Therefore, a dynamic emotional intensity modeling module (detailed in Section 3.3) is proposed to extract this information.

## 3 METHOD

**Problem Formulation.** Let $C_{1:T} = (c_1, \ldots, c_T)$, $c_t \in \mathbb{R}^{185}$ be a $T$-length sequence of control rig coefficients, where each frame $c_t$ has 185 control rig data. Also, we define $A_{1:T} = (a_1, \ldots, a_T)$ as a sequence of speech snippets, each of which $a_t \in \mathbb{R}^d$ has $d$ samples to align with the corresponding visual frame $c_t$. Moreover, $e^{sty} \in \mathbb{R}^5$ denotes a user-controllable emotional style which is

one-hot encoding. Then, our goal is to predict emotion-enhanced coefficients $\hat{C}_{1:T} = (\hat{c}_1, \ldots, \hat{c}_T)$ conditioned on $A_{1:T}$ and $e^{sty}$.

### 3.1 Overview

The whole pipeline of our approach for the speech-driven 3D emotional facial animation task is revealed in Figure 2. To address existing limitations, such as lack or improper handling of emotional intensity modeling, we carefully design two modules: the Dynamic Emotional Intensity Modeling Module (DEI, detailed in Section 3.3) and the Emotion-guided Feature Fusion Decoding Module (detailed in Section 3.4). The former extracts high time-frequency representations of emotional intensity from speech features, while the latter maps audio to facial coefficients using emotional information derived from emotional style input and DEI. Moreover, we design a dynamic positional encoding (DPE) strategy to provide temporal-order information for speech sequences of varying lengths. In summary, DEITalk takes input with a speech sequence $A_{1:T}$ and an emotional style label $e^{sty}$, and produces emotion-enhanced control rig coefficients $\hat{C}_{1:T}$. These predicted coefficients can be utilized to animate MetaHuman characters, enhancing their emotional expression. Formally, we define our DEITalk as:

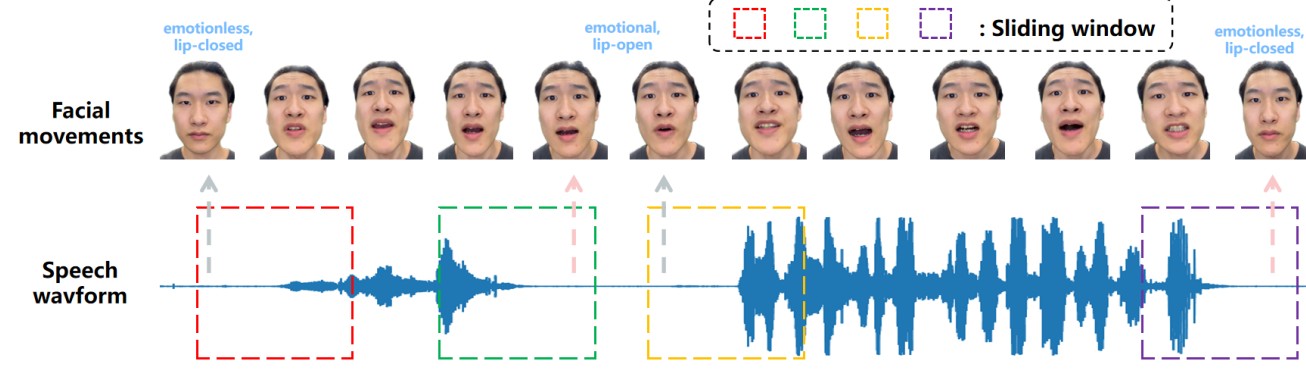

Figure 3: Problems caused by using sliding windows. When emotional speech is segmented using sliding windows, the periods of silence within the window may correspond to different facial expressions. For instance, when the sliding window is positioned at the beginning or end of the speech sequence, the silence within it should correspond to a neutral expression with closed lips. However, when the sliding window is placed in the middle of the original speech sequence, the periods of silence are more likely to correspond to expressions with subtle emotions (influenced by the long-term dependencies of emotions) and slightly open lips (mainly influenced by breathing).

$$\hat{\mathbf{C}}_{1:T} = \mathrm{DEITalk}_\theta(\mathbf{A}_{1:T}, \mathbf{e}^{sty}), \qquad (1)$$

where $\theta$ indicates the model parameters. In the following sections, we will introduce the details of our method.

## 3.2 Audio Processing

**Audio Feature Extraction.** Before feeding the raw speech signal $\mathcal{X}$ to our model, it will be converted into a speech sequence $\mathbf{A}_{1:T}$ via the Librosa library [25] at a sampling frequency of 16kHz. Then, we employ Wav2Vec 2.0 [2] to encode $\mathbf{A}_{1:T}$, which represents the state-of-the-art self-supervised pre-trained speech model for ASR [41] task. Wav2Vec 2.0 consists of several temporal convolutional layers and transformer encoder layers, which, respectively, convert the audio input into feature vectors sampled at 50Hz and further encode them into contextualized speech representations $\mathbf{S}_{1:T} = (\mathbf{s}_1, \ldots, \mathbf{s}_T)$, $\mathbf{s}_t \in \mathbb{R}^{768}$. Formally, it can be defined as $\mathcal{E}_A(\mathbf{A}_{1:T}) \rightarrow \mathbf{S}_{1:T}$, where $\mathcal{E}_A$ is the Wav2Vec2.0 architecture. In addition, a linear interpolation layer is applied to resample the speech feature at 30Hz to match the frequency $f_c$ of the control rig coefficients. Moreover, to maximize data utilization, we employ large overlap sliding windows to segment speech sequences during the training phase.

**Dynamic Positional Encoding.** Facial movements for emotional expression exhibit longer temporal dependencies. Utilizing sliding windows for speech segmentation could potentially result in the loss of positional information regarding the original complete speech sequences (depicted in Figure 3). Thus, endeavoring to reconstruct precise emotional expressions exclusively from speech segments may introduce ambiguity within the system. Additionally, the sinusoidal position encoding method employed in transformer architectures [36] exhibits limited generalization capabilities [8] when confronted with sequences longer than those encountered during the training phase.

To address this issue, we introduce a dynamic positional encoding (DPE) strategy that can effectively handle sequences of different

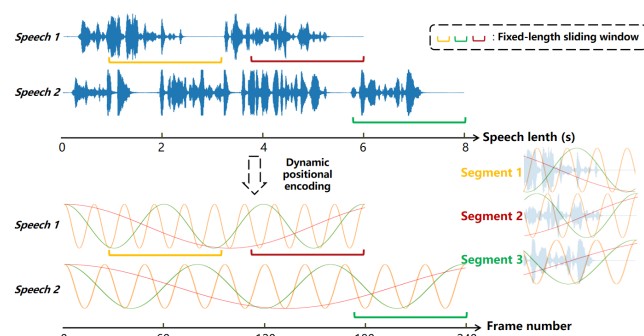

Figure 4: Mechanism of DPE. We capture three equally sized segments from two speech sequences of different lengths using sliding windows. Segments 2 and 3 are both located at the tail, which means their corresponding facial expressions should be more similar during the silence, while segment 1 is located in the middle, meaning its facial expressions during the silence should be more different from those of segments 1 and 2. Although these three segments share very similar waveform characteristics, DPE effectively provides positional information relative to the original speech sequences(e.g., despite being located at different indices within their respective sequences, segments 1 and 2 exhibit a high degree of similarity in their DPE results).

lengths and describe the positional information of each time step relative to the complete sequence. Specifically, we modify the original sinusoidal positional encoding method to enable adaptive position encoding tailored to speech sequences of varying lengths $T$:

$$
\begin{aligned}
\mathbf{DPE}_{(t,2i)} &= \sin\left(\frac{20\pi \cdot t}{T \cdot 10000^{2i/d}}\right) \\
\mathbf{DPE}_{(t,2i+1)} &= \cos\left(\frac{20\pi \cdot t}{T \cdot 10000^{2i/d}}\right)
\end{aligned}
\qquad (2)
$$

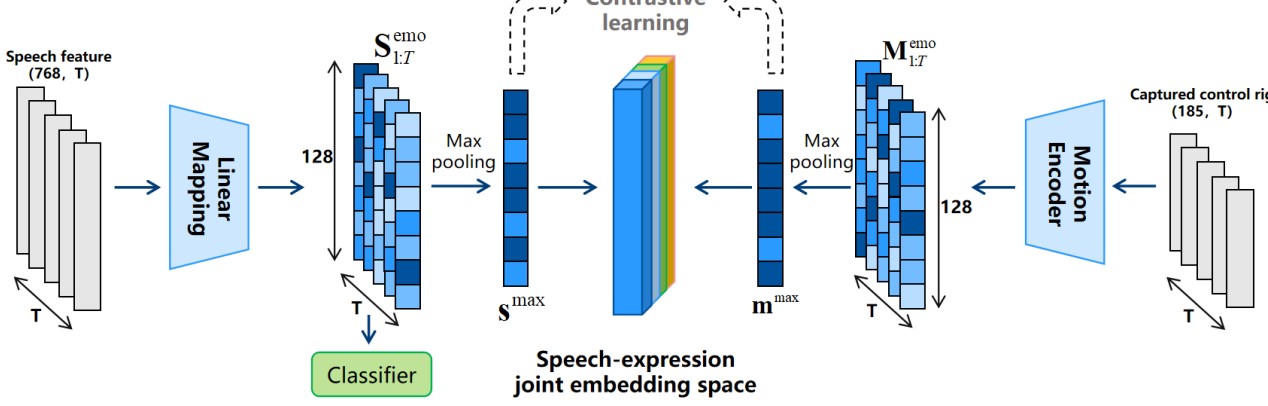

**Figure 5: We learn a speech-expression joint embedding space using contrastive learning. The speech feature sequence is converted to a compressed sequence $S_{1:T}^{emo}$, which is then aggregated into an embedding vector $s^{max}$ via max pooling. Similarly, the captured control rig coefficients are processed by a motion encoder, producing a feature sequence $S_{1:T}^{emo}$ and the corresponding embedding vector $m^{max}$. To ensure that $S_{1:T}^{emo}$ can capture pertinent emotional information, a classifier is introduced to predict emotional categories. This module is trained using a contrastive loss that maximizes the similarity between the embeddings $s^{max}$ and $m^{max}$ of paired speeches and control rig coefficients.**

where $t$ is the current time step in the sequence, $d$ is the dimension of the model, and $i$ is the dimension index. Instead of assigning a unique position identifier for each time step, the proposed DPE strategy injects position information into each time step relative to the entire sequence (as shown in Figure 4). DPE is applied to the speech features $S_{1:T}$ to provide temporal order information. For each feature vector $s_t$ in $S_{1:T}$:

$$\hat{s}_t = s_t + \mathbf{DPE}(t, T). \tag{3}$$

## 3.3 Dynamic Emotion Intensity Modeling

In light of the potential limitations associated with employing one-hot vectors to characterize the intensity of facial movements, our goal is to learn an implicit representation of emotional intensity from the extracted speech features and to represent it as a spatial-local, high-temporal-frequency feature. This feature serves as one of the conditions that guide the decoder in synthesizing facial animation with rich expressions.

**Speech-Expression Joint Embeddings.** The many-to-many mapping between speech content and facial expression challenges in generating semantically correct facial movements. Inspired by the Contrastive-Language-Image-Pretraining (CLIP) model [30] and GestureDiffuCLIPmodel [1], we integrate contrastive learning with a temporal aggregation mechanism to establish a joint embedding space for speech and facial movements in emotional expression. This space offers emotional saliency cues that steer the system toward learning the correspondence of emotional intensity across these two distinct modalities.

**Architecture.** As shown in Figure 5, a linear projection layer is applied to map the speech feature $S_{1:T}$ into a lower-dimensional feature $S_{1:T}^{emo} = (s_1^{emo}, \ldots, s_T^{emo})$, $s_t^{emo} \in \mathbb{R}^{128}$. Meanwhile, we designed a motion encoder $\mathcal{E}_M$ that comprises a four-layer TCN along with a linear layer to extract the ground truth of the control rig coefficients $C_{1:T}$, which can be defined as $\mathcal{E}_M(C_{1:T}) \rightarrow M_{1:T}^{emo} = (m_1^{emo}, \ldots, m_T^{emo})$, $m_t^{emo} \in \mathbb{R}^{128}$. To mitigate confusion

during the training phase caused by incomplete temporal alignment between these two modalities, we employ maximum pooling to aggregate emotion-relevant information from each feature sequence:

$$s^{max} = \text{max\_pooling}\left(S_{1:T}^{emo}\right),$$
$$m^{max} = \text{max\_pooling}\left(M_{1:T}^{emo}\right). \tag{4}$$

Then, $s^{max}, m^{max}$ are considered the embeddings of speech and facial movements, respectively. To avoid misinterpreting speech-induced articulation for emotional cues, we introduce a classifier to predict the emotion style $\hat{e}^{sty}$ from $S_{1:T}^{emo}$ and compare it with the label of the emotional style of the ground truth $e^{sty}$.

**Contrastive Learning.** We apply CLIP-style contrastive learning to finetune the module. Specifically, given a batch of pairs of embeddings $\mathcal{N} = \{(s_i^{max}, m_i^{max})\}_{i=1}^N$, where $N$ is the batch size, our training goal is to maximize the similarity of the embeddings $(s_i^{max}, m_i^{max})$ of the real pairs while minimizing the similarity of the incorrect pairs $(s_i^{max}, m_j^{max})_{i \neq j}$.

**Applications of the Joint Embeddings.** The joint embedding space provides an effective method to measure the similarity of emotional saliency between speech and facial movements, thereby enabling the characterization of dynamic emotional intensity. Given $S_{1:T}^{emo}$ and its embedding vector $s^{max}$, we calculate the similarity between $s_t^{emo}$ and $s^{max}$, and obtain the corresponding emotional saliency sequence $E_{1:T}^{sim} = (e_1^{sim}, \ldots, e_T^{sim})$, $e_t^{sim} \in \mathbb{R}$:

$$e_t^{sim} = \cos(s_t^{emo}, s^{max}), \ t \in \{1, 2, \ldots, T\}. \tag{5}$$

As shown in Figure 6, frames with high similarity scores are more likely to accompany intense emotional expressions. These cues serve as significant emotional saliency indicators and will be leveraged within our system to direct the generator in synthesizing expressive facial animations. Then, the desired dynamic emotional intensity representations $E_{1:T}^{in} = (e_1^{in}, \ldots, e_T^{in})$, $e_t \in \mathbb{R}^{128}$ can be acquired by element-wise multiplication of $S_{1:T}^{emo}$ and $E_{1:T}^{sim}$:

$$e_t^{in} = e_t^{sim} s_t^{emo}, \ t \in \{1, 2, \ldots, T\}. \tag{6}$$

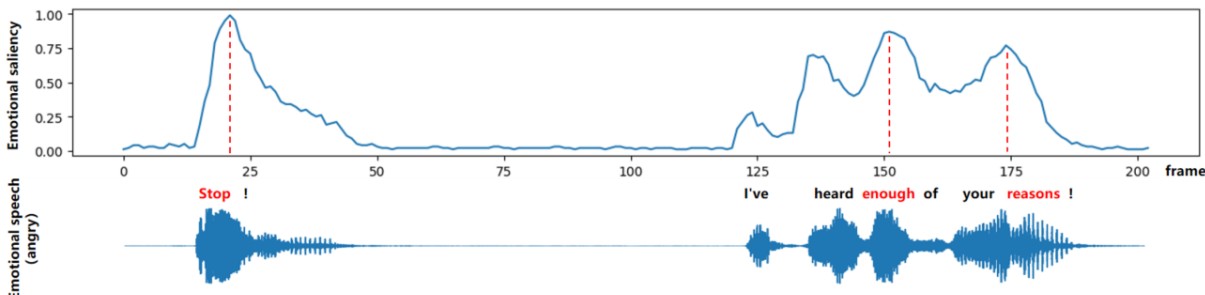

**Figure 6: Visualization of emotional-saliency curves. The peak of the curve represents frames with high emotional semantic importance, which may be accompanied by intent expressions.**

## 3.4 Emotion-Guided Animation Generation

We propose an emotion-guided feature fusion decoder, where emotion consists of two key components: emotional style embedding (controlled by the user) and dynamic emotional intensity representations (extracted by DEI). By effectively leveraging these emotional cues, the proposed decoder aims to map the speech to emotional 3D facial animation coefficients.

Previous studies implement the decoder adhering to the structure of the transformer decoder [10, 28]. However, the transformer decoder requires autoregressive output, meaning that each frame's output is contingent upon the decoding of all speech features. Predicting facial movements for long-context speech would result in an undesirable increase in time complexity, especially in scenarios demanding swift inference. Given the inherent constraints of the transformer decoder, we employ an eight-layer TCN to construct our decoder and meticulously regulate the receptive field of each layer to manage the influence of each frame on its neighborhood.

The one-hot emotional style embedding is first mapped to a 16-dimensional vector $E^{sty}$ and then padded to match the length of the speech feature $S_{1:T}$. Consequently, the speech feature $S_{1:T}$ is concatenated with $E^{sty}_{1:T}$ and $E^{in}_{1:T}$ to get the fused feature $F$. Then, $F$ will be fed into our decoder. In addition, we also used a similar feature fusion strategy in each temporal convolutional layer of TCN, the overall formula for feature fusion can be described as:

$$F_i = \begin{cases} \text{Concatenate}(S_{1:T}, E^{sty}_{1:T}, E^{in}_{1:T}), & i = 1 \\ \text{Concatenate}(F_{i-1}, E^{sty}_{1:T}, E^{in}_{1:T}), & i \in \{2, 3, \dots, 8\} \end{cases}. \quad (7)$$

Finally, a fully connected layer is employed to map the outputs of the emotion-guided feature fusion decoder to 185-dimensional control rig coefficients.

## 3.5 Objective Function

We design a novel loss function comprising four distinct components: contrast loss, classification loss, position loss, and velocity loss. The overall function is given by:

$$\mathcal{L} = \lambda_1 \mathcal{L}_{con} + \lambda_2 \mathcal{L}_{cla} + \lambda_3 \mathcal{L}_{pos} + \lambda_4 \mathcal{L}_{vel}, \quad (8)$$

where we set the the trade-off parameters with the values as follows $\lambda_1 = 0.1$, $\lambda_2 = 0.1$, $\lambda_3 = 1.0$, and $\lambda_4 = 4.0$ by default in our experiments. In the following sections, we will provide a detailed explanation of each component.

**Contrast Loss.** The contrast loss function is designed to optimize the alignment between speech and facial movements while learning the joint embedding space. Given a batch of samples with a batch size of $N$, let $s^{max}_i$ be the vector representation of speech $i$ and $m^{max}_j$ be the vector representation of facial movements $j$. Consequently, we obtain the similarity score $s_{ij}$ by computing the cosine similarity between these two vectors:

$$s_{ij} = \cos(s^{max}_i, m^{max}_j), \quad (9)$$

Subsequently, the contrast loss function can be expressed as:

$$\mathcal{L}_{con} = -\frac{1}{2N} \sum_{i=1}^{N} \left[ \log \frac{\exp(s_{ii}/\tau)}{\sum_{j=1}^{N} \exp(s_{ij}/\tau)} + \log \frac{\exp(s_{ii}/\tau)}{\sum_{j=1}^{N} \exp(s_{ji}/\tau)} \right], \quad (10)$$

where $\tau$ is the temperature parameter that adjusts the scaling of the similarity scores.

**Classification Loss.** To ensure that the learned joint embedding space can sufficiently describe emotional intensity, we introduce a classification loss to constrain the learning process of our system. The classification loss is defined as:

$$\mathcal{L}_{cla} = -\frac{1}{N} \sum_{i=1}^{N} \sum_{c=1}^{M} y^{sty}_{ic} \log(p^{sty}_{ic}), \quad (11)$$

where $M$ denotes the number of classifications, $y^{sty}_{ic}$ is the observation function that determines whether the sample $i$ carries the emotion label $c$, and $p^{sty}_{ic}$ denotes the predicted probability that sample $i$ belongs to class $c$.

**Position Loss.** The position loss computes the distance between the predicted outputs and the ground truth of control rig sequences, which is described as:

$$\mathcal{L}_{pos} = \frac{1}{N} \frac{1}{T} \sum_{i=1}^{N} \sum_{t=1}^{T} \left\| c^i_t - \hat{c}^i_t \right\|^2_2, \quad (12)$$

where $T$ represents the length of each sample in the training set (i.e., the window size of the sliding window), $c^i_t$ and $\hat{c}^i_t$ denote the ground truth and predicted values, respectively, for the $t$-th frame of sample $i$. The position loss serves to encourage the model to match the ground-truth performance.

**Velocity Loss.** The velocity loss computes the distance between the differences of consecutive frames between predicted outputs and the ground truth of control rig sequences, inducing the generation with temporal stability. The formula is as follows:

$$\mathcal{L}_{vel} = \frac{1}{N} \frac{1}{T-1} \sum_{i=1}^{N} \sum_{t=1}^{T-1} \left\| \left( c^i_{t+1} - c^i_t \right) - \left( \hat{c}^i_{t+1} - \hat{c}^i_t \right) \right\|^2_2. \quad (13)$$

# 4 EXPERIMENTS

## 4.1 Experimental Settings

**Datasets.** We train and validate DEITalk on our dataset, ETH-3D. ETH-3D is an audio-visual dataset consisting of five different styles of emotions (i.e., Neutral, Angry, Happy, Sad, and Surprised), and each emotional style corresponds to approximately 100 samples, with each sample lasting between 8 and 15 seconds. Specifically, the emotional intensity of each sample dynamically changes in accordance with the semantic salience of the corresponding transcribed text. We split the dataset with 70% of the samples included in the training set (ETH-3D-Train), 20% in the validation set (ETH-30-Val), and 10% in the testing set (ETH-3D-Test). For a broader evaluation, a small-scale, multilingual test dataset was created for generalization testing, where the speakers of the audio sequences are disjoint from those in ETH-3D.

**Network Architectures.** In the dynamic emotional intensity modeling module, the TCN within the motion encoder consists of four 1D convolutional layers, each employing the ReLU activation function. The configuration includes {384, 384, 768, 768} filters for the respective layers, all with a filter size of 3, a stride of 1, and dilation and padding values set to {1, 2, 4, 8}, respectively. The classifier comprises two fully connected layers, yielding output channels of {64, 5}. In the emotion-guided feature fusion decoding module, the TCN within the decoder incorporates eight 1D convolutional layers with ReLU activation. Each layer is equipped with 512 filters, all set with a filter size of 3, a stride of 1, and dilation and padding values of {1, 2, 4, 8, 1, 2, 4, 8}, respectively.

**Implementation Details.** Our system generates facial parameters represented by a 185-dimensional control rig at a rate of 30 frames per second. For the overall training procedure, we employ Adam as optimizer, with a batch size of 32 and a learning rate set to 1e-4. Speech sequences are segmented using a sliding-window approach, with a stride of 5 frames and a window size of 128 frames. The TCN parameters in Wav2Vec2.0 are fixed, whereas the other components of the model are learnable. The value of $\tau$ in contrast loss is set to 0.07. DEITalk is implemented using PyTorch and takes around 10 hours (300 epochs) to train the whole network on ETH-3D with a single NVIDIA RTX3090 GPU.

**Baseline Methods.** We compare DEITalk with two state-of-the-art open-source methods, FaceFormer [10] and EmoTalk [28], on ETH-3D-Test and wild audio, respectively, and render the predicted animations on the same MetaHuman avatar. Notably, FaceFormer requires a neutral 3D face mesh template during both the training and inference stages, with its output being the mesh vertices of the face. Since these elements are unavailable in our dataset, we made necessary adjustments to its network structure. This ensures that it can be trained on our dataset and facilitates a more equitable comparison with our proposed method.

## 4.2 Qualitative Evaluation

Lip movement and emotional expression are two crucial aspects in assessing the quality of facial animation. For instance, the lip vertex error (LVE) is used in MeshTalk [31] and FaceFormer [10] to measure the quality of lip synchronization, while the emotional vertex error (EVE) is used in EmoTalk [28] to reflect the quality of

**Table 1: Quantitative evaluation results on ETH-3D-Test.**

| Method | LPE↓ | EPE↓ |
|---|---|---|
| FaceFormer [10] | 0.00353 | 0.00434 |
| EmoTalk [28] | 0.00194 | 0.00128 |
| **Ours** | **0.00162** | **0.00079** |

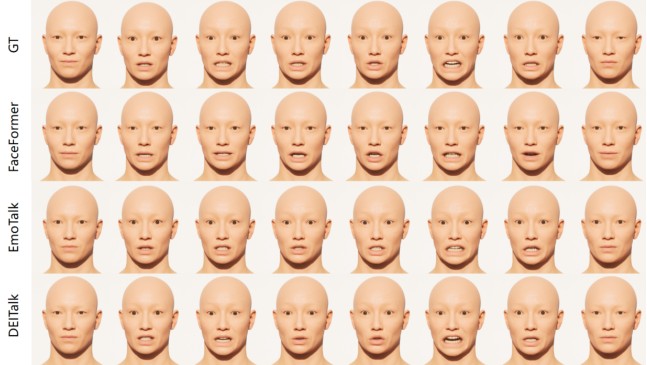

**Figure 7: Quantitative evaluation results of the rendered animation on ETH-3D-Test. We compare our results with the SOTA methods in a "surprised" emotional style.**

emotional expression. However, as LVE and EVE calculate distance errors of facial vertices, they are not directly applicable to evaluate DEITalk. This is because DEITalk operates as a parameter-based model that cannot directly output facial vertex information. To overcome this limitation, we draw inspiration from LVE and EVE and introduce a new set of evaluation metrics: lip parameter error (LPE) and emotion parameter error (EPE).

**Lip Parameter Error.** LPE measures the maximum error $\ell_2$ between predicted values and ground truth for 86 specific control rig coefficients, which are responsible for controlling the movements of the lip and its surrounding areas.

**Emotional Parameter Error.** Similar to LVE, EPE measures the maximum error $\ell_2$ between predicted values and ground truth for 18 specific control rig coefficients governing the movements of the eyes, eyebrows, forehead, and surrounding areas.

We report the LPE and EPE evaluation results for FaceFormer, EmoTalk, and DEITalk in Table 1. Compared to the other two methods, DEITalk achieved a lower lip error and emotion expression error, indicating that it can produce more accurate lip movements and more realistic emotional expressions.

## 4.3 Qualitative Evaluation

Due to the many-to-many mapping nature of the audio-to-face task, solely relying on metrics for qualitative evaluation is insufficient. Human perceptual assessment is also needed. Consequently, we proceed to qualitatively evaluate our model from two perspectives.

**Lip Synchronization.** We compared our model with FaceFormer and EmoTalk by feeding them identical audio inputs and generating corresponding facial animations. To avoid introducing emotion-related interference in the lip synchronization evaluation, all audio inputs belong to the neutral emotion style. The comparison results (shown in our supplementary video) show that DEITalk exhibits

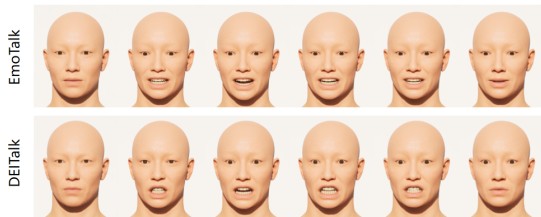

Figure 8: Quantitative evaluation results of the rendered animation on wild speech. Given a wild speech with an emotional style of anger, EmoTalk extracts wrong emotional information and synthesizes an animation expressing joy. In contrast, DEITalk generates results that are more congruent with the emotional style of the speech.

Table 2: User study results.

| Method | FaceFormer | Ours |
|---|---|---|
| full-face | 31.9% | **68.1%** |
| lip sync | 39.6% | **60.4%** |
| emotion expression | 27.2% | **72.8%** |
| Method | EmoTalk | Ours |
| full-face | 35.4% | **64.6%** |
| lip sync | 43.7% | **56.3%** |
| emotion expression | 31.5% | **68.5%** |

more accurate lip movements and better aligns with human pronunciation patterns. Moreover, DEITalk exhibited robust generalization capabilities even in zero-shot cases featuring different languages.

**Emotional Expression.** We feed FaceFormer, EmoTalk, and DEITalk identical audio inputs containing various emotional styles and compare the emotional variation in the facial animations generated by them (detailed in our supplementary video). Figure 7 shows a comparison result of a specific emotional style, the animations generated by FaceFormer exhibit limited emotional differentiation due to the lack of optimization for emotional expression. EmoTalk and DEITalk exhibit comparable performance on the ETH-3D-Test, with DEITalk slightly ahead in the realism of emotional variations. In zero-shot settings, EmoTalk exhibits severely limited emotional expression capabilities (shown in Figure 8), often resulting in emotionless or even erroneous emotional output. On the contrary, DEITalk maintains a relatively superior and robust emotional expression performance. This is primarily attributed to the fact that DEITalk treats the emotional style as a user-controllable condition and dynamically analyzes the emotional intensity of the input audio.

We strongly recommend that the reader watch our supplementary video, which offers more detailed comparisons to assess the motion quality of our approach.

### 4.4 User Study

To comprehensively evaluate the performance of our proposed model, we conducted an extensive user study with a total of 136 participants. The evaluation process was divided into three subtasks: full-face comparison, lip-sync comparison (by covering the area above the nose), and emotion expression comparison (by covering the area below the nose). We selected 20 test cases (four for each

Table 3: Ablation study for our components.

| Strategy | LPE↓ | EPE↓ |
|---|---|---|
| **Ours** | **0.00162** | **0.00079** |
| w/o DEI modeling module | 0.00220 | 0.00247 |
| w/o dynamic positional encoding | 0.00188 | 0.00086 |
| w/o $\mathcal{L}_{vel}$ loss | 0.00373 | 0.00108 |
| w/o $\mathcal{L}_{cla}$ loss | 0.00182 | 0.00094 |

emotional style), half of which are from ETH-3D, while the other half are from wild settings, resulting in ultimately 120 choice questions. Participants were required to choose more realistic videos from these one-to-one comparison videos.

Table 2 reports a detailed description of the user selection results. Our method achieved the highest positive evaluation results in all three sub-tasks, notably excelling in the emotion expression comparison task, where participants considered our approach significantly superior to others.

### 4.5 Ablation Experiment

To investigate the contributions of the key designs of our proposed method, including the dynamic positional encoding strategy, dynamic emotional intensity modeling module, and loss function, we conducted an ablation experiment using the evaluation metrics introduced in Section 4.2.

As shown in Table 3, the removal of the dynamic emotional intensity modeling module results in a significant increase in EPE, demonstrating its crucial role in the representation of emotional intensity. Meanwhile, abandoning the dynamic positional encoding strategy results in a slight increase in LPE and EPE, which indicates the effectiveness of DPE in enhancing emotional expression.

From the perspective of the loss function, removing velocity loss leads to a clear decrease in performance, inducing noticeable shaking in the facial animation output. Removing the classification loss significantly increased the EPE, indicating that the dynamic emotional intensity modeling module could no longer effectively express emotional information.

## 5 CONCLUSION

In this paper, we propose a novel speech-driven 3D facial animation framework (DEITalk) that achieves state-of-the-art performance in the generation of emotional 3D talking heads with aligned lip movements and realistic expressions. Unlike previous methods that regarded emotional intensity as a discrete label (e.g., one-hot encoding), we specifically design a dynamic emotional intensity modeling mechanism that allows the model to automatically analyze high-frequency emotional intensity information from speech features and provide reliable emotional saliency guidance for facial animation generation. To further enhance the emotional expression ability of DEITalk, we introduce a dynamic positional encoding strategy, an emotion-guided feature fusion decoder, and a four-way loss function, respectively. Moreover, DEITalk is a lightweight and artist-friendly model whose outputs are identity-free and can be directly utilized to manipulate facial animations of any MetaHuman avatar, making it seamlessly integrated into industrial pipelines.

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
