# OpenReview forum: "DEITalk: Speech-Driven 3D Facial Animation with Dynamic Emotional Intensity Modeling"
_acmmm.org/ACMMM/2024/Conference — MM2024 Poster_

### Official Review · Reviewer_oTUb · 2024-05-12

**Rating:** 4
**Confidence:** 4

**Summary:**

This paper presents DEITalk, a novel approach for speech-driven 3D facial animation. It emphasizes the importance of modeling emotional intensity and introduces the Dynamic Emotional Intensity (DEI) module to learn this from speech. The model is guided by both the DEI module and an emotional style label to generate emotive 3D facial animations. The authors collect a new 3D emotional talking head dataset, ETH-3D, to train and evaluate their model.

**Strengths:**

1. This paper proposes a approach, DEITalk, for speechdriven 3D facial animation, which enables the synthesis of expressions with a specified emotional style while adapting to the dynamic emotional intensity in the speech.
2. They introduce a dynamic emotional intensity modeling module that employs a contrastive learning strategy to learn the correlation between speech and facial movements in emotional saliency.
3. They capture a 3D emotional talking head dataset, ETH- 3D, comprising five distinct emotional styles.

**Limitations:**

1. The DEITalk shown in the supplementary video seems to work well for Chinese, but poor lip synchronization for other languages.
2. Will ETH-3D be made public? If not, it will contribute less to the community. What is the total duration of ETH-3D? The description in 4.1 seems to be a little short.
3. In Fig 3, the sliding windows currently used are generally not as large as the picture. For example, it is 0.2 seconds in wav2lip, and emotion does not change much.
4. On line 382, wav2vec 2.0 is not yet the state-of-the-art, you can try using hubert.
5. In Figure 6, the emotional-saliency curves appear to correlate with volume, while no significant change in emotional intensity is seen in the supplementary video.
6. There are few methods of comparison, and it is recommended to include evaluations for CodeTalker and SelfTalk.

**Suitability:**

3

---

### Official Review · Reviewer_ktFD · 2024-05-19

**Rating:** 2
**Confidence:** 4

**Summary:**

This paper first collected a 3D talking head dataset comprising five emotional styles with a set of coefficients based on the MetaHuman character model. Next, the paper proposed an end-to-end deep neural network, DEITalk, which conditions on speech and emotional style labels to generate realistic facial animation with dynamic expressions. Extensive experimental results demonstrate that their method outperforms existing state-of-the-art methods.

**Strengths:**

- The paper presents an emotional 3D talking head dataset, which contributes to the development of the field of Speech-Driven 3D Facial Animation.
- The paper recognizes the shortcomings of the global emotion modeling using one-hot encoding and proposes a dynamic emotional intensity module to achieve fine-grained emotional expression.

**Limitations:**

- The dataset proposed in this paper is relatively small, which limits its practical applications to some extent.
- All experiments in this paper were conducted on the ETH-3D dataset, without comparisons on other public datasets (e.g., EmoTalk[1]). Therefore, the persuasiveness of the experimental results is limited. Moreover, sometimes better results may be attributed to the use of a superior dataset, especially compared to using more facial coefficients than the 52-dimensional blendshape coefficients.
- Recently, methods based on diffusion models, such as FaceDiffuser[2], have shown significant advantages in this field. The paper should include comparative experimental results to validate the advanced nature of the proposed model.
- In the speech-expression joint embedding space, the captured control rig (185,T) contains both emotional and semantic information. Therefore, is it reasonable to achieve fine-grained emotional alignment through this alignment method?
- From the visual effects of the demo video, one can observe the overall emotional changes, but it is difficult to discern the frame-level fine-grained emotional changes, which raises doubts about the effectiveness of the model.


[1] Peng Z, Wu H, Song Z, et al. Emotalk: Speech-driven emotional disentanglement for 3d face animation. In Proceedings of the IEEE/CVF International Conference on Computer Vision. 2023: 20687-20697.

[2] Stan S, Haque K I, Yumak Z. Facediffuser: Speech-driven 3d facial animation synthesis using diffusion. In Proceedings of the 16th ACM SIGGRAPH Conference on Motion, Interaction and Games. 2023: 1-11.

**Suitability:**

3

---

### Official Review · Reviewer_aBU4 · 2024-05-25

**Rating:** 4
**Confidence:** 4

**Summary:**

This paper introduces a novelty method for generating speech-driven 3D facial animations, focusing on dynamic emotional intensity modeling. The authors create a new dataset consisting of five emotional styles with a set of coefficients based on the MetaHuman character model. The DEITalk is designed to model emotional saliency variations in long-term audio contexts with a dynamic emotional intensity (DEI) modeling module and a dynamic positional encoding (DPE) strategy.

**Strengths:**

1. This paper presents a new dataset and an end-to-end deep neural network, DEITalk, to synthesize 3D talking head animations with precise lip movements and rich stylistic expressions.
2, The paper is well-structured, starting with a clear identification of the limitations in existing speech-driven 3D facial animation methods, particularly their lack of emotional expression and the over-smoothing of emotional intensity. The authors effectively argue that incorporating dynamic emotional intensity modeling can significantly enhance the realism of 3D facial animations.
3. The experimental demos presented in the paper clearly demonstrate the effectiveness of emotion control. The DEITalk method produces animations with dynamic emotional expressions that vary in intensity over time, which is visually evident in the results. These animations are compared with state-of-the-art methods, and DEITalk consistently shows superior performance.

**Limitations:**

1. It is unclear whether the ETH-3D dataset introduced in the paper will be made publicly available. Without open access, the dataset cannot be considered a significant contribution as it does not offer novel data processing procedures or insights into how different dataset characteristics affect model performance (e.g., scaling laws).
2, The pipeline for data collection using MetaHuman and LiveLink Face, which is used for animation generation, has already been validated in prior work such as “Transformer-S2A: Robust and Efficient Speech-to-Animation.” Therefore, this aspect of the paper does not present a novel contribution to the field.
3. The Dynamic Positional Encoding (DPE) introduced in the paper lacks references to FaceFormer’s Periodic Positional Encoding (PPE). The paper does not clarify the connection or distinctions between DPE and PPE, nor does it provide a justified motivation for introducing DPE over existing methods.
4. Section 4.5 does not provide an evaluation demonstrating that DPE outperforms the sinusoidal position encoding used in Transformers or FaceFormer’s PPE. Moreover, the demo video does not include an ablation study focusing on positional encoding.
5. The demo videos show noticeable emotional changes in the mouth corners and eye areas, but the cheek region appears to remain mostly static. This does not align with how human faces typically display emotions, suggesting a limitation in the model’s ability to generate comprehensive facial expressions.
6. Although the paper claims to learn continuous emotional intensity from speech, the experimental section does not detail how different intensities affect the animation. Emotion is still treated as a global variable, which is inconsistent with the paper’s motivation. To address this, the paper should include tests showing the impact of varying intensities for the same sentence on animation, such as using the MEAD dataset for evaluation.

**Suitability:**

3

---

### Meta-Review · Area_Chair_19PX · 2024-07-01

**Recommendation:** Accept (Poster)
**Confidence:** 4

**Metareview:**

This paper introduces a novelty method for generating speech-driven 3D facial animations, focusing on dynamic emotional intensity modeling. After the authors' rebuttal, Reviewers found an agreement on considering the paper contribution valuable for publication.  So I tend towards accepting this paper as a poster.